# Multiplex Real-Time PCR Assay for Six Major Carbapenemase Genes

**DOI:** 10.3390/pathogens10030276

**Published:** 2021-03-01

**Authors:** Nori Yoshioka, Hideharu Hagiya, Matsuo Deguchi, Shigeto Hamaguchi, Masanori Kagita, Isao Nishi, Yukihiro Akeda, Kazunori Tomono

**Affiliations:** 1Division of Infection Control and Prevention, Osaka University Hospital, 2-15 Yamadaoka, Suita, Osaka 565-0871, Japan; hagiya@okayama-u.ac.jp (H.H.); exit@hp-lab.med.osaka-u.ac.jp (M.D.); hamaguchi@hp-infect.med.osaka-u.ac.jp (S.H.); kagita@hp-lab.med.osaka-u.ac.jp (M.K.); nishi@hp-lab.med.osaka-u.ac.jp (I.N.); akeda@biken.osaka-u.ac.jp (Y.A.); tomono@hp-infect.med.osaka-u.ac.jp (K.T.); 2Laboratory for Clinical Investigation, Osaka University Hospital, 2-15 Yamadaoka, Suita, Osaka 565-0871, Japan; 3Department of General Medicine, Okayama University Graduate School of Medicine, Dentistry and Pharmaceutical Sciences, 2-5-1 Kitaku Shikatachou, Okayama 700-8558, Japan

**Keywords:** carbapenem-resistant *Enterobacteriaceae*, carbapenemase-producing *Enterobacteriaceae*, infection control, multiplex detection assay

## Abstract

The global dissemination of carbapenemase-producing *Enterobacteriaceae* (CPE) is a major concern in public health. Due to the existence of the diversity of carbapenemases, development of an easily available, cost-effective multiplex detection assay for CPE is required worldwide. Using clinically available and reliable equipment, COBAS^®^ z480 (Roche Diagnostics K.K., Tokyo, Japan), we developed a multiplex real-time PCR assay for the detection of two combinations of carbapenemases; first, *bla*_NDM_, *bla*_KPC_, and *bla*_IMP_ (Set 1), and second, *bla*_GES_, *bla*_OXA-48_, and *bla*_VIM_ (Set 2). We constructed standard curves for each carbapenemase gene using serial dilutions of DNA standards, then applied reference or clinical isolates with each carbapenemase gene to this assay. The multiplex assay showed satisfactory accuracy to detect CPE genes, with the correlation coefficients of greater than 0.99 for all genotypes. The assay appropriately differentiated the reference or clinical strains harboring each carbapenemase gene without cross reactivity. Lastly, the assay successfully detected multiple genes without false-positive reactions by applying six clinical isolates carrying both NDM and OXA-48-like carbapenemase genes. Major advantages of our assay include multiplicity, simple operation, robustness, and speed (1 h). We believe that the multiplex assay potentially contributes to early diagnosis of CPE with a diverse genetic background.

## 1. Introduction

The high antimicrobial resistance of carbapenemase-producing *Enterobacteriaceae* (CPE) is a significant threat to global public health [1,2]. Carbapenemases are classified into two groups; (i) metallo-β-lactamases (MBLs), such as NDM, VIM, and IMP, and (ii) serine-β-lactamases, such as KPC, OXA-48, and GES. Due to this genetic diversity, it is challenging to appropriately detect and differentiate CPE isolates at in-house laboratories.

To date, IMP-type CPE strains are predominantly found in Japan [3,4]; however, the risk of importing other CPE strains also increases with globalization. In this context, the development of a multiplex assay for several CPE strains is required to prevent and control the spread of infection. Several assays for the detection of CPE isolates have been developed [5,6,7], including immunochromatographic tests [8,9,10]. Here, we describe a novel multiplex assay to detect six major carbapenemase genes using previously developed single PCR primers and a clinically available COBAS^®^ z480 (Roche Diagnostics K.K., Tokyo, Japan) equipment in a hospital setting.

## 2. Materials and Methods

### 2.1. Laboratory Examination

#### 2.1.1. Bacterial Strains

We then applied the following reference or clinical isolates of each carbapenemase gene to this assay: NDM, a clinical strain of *bla*_NDM-5_-positive *Klebsiella pneumoniae* (accession numbers DRX117870 and DRX117871); KPC, an ATCC reference strain of *K. pneumoniae* BAA-1705 possessing *bla*_KPC-2_ (accession number AOGQ00000000); IMP, a clinical strain of *bla*_IMP-6_ positive *K. pneumoniae* (accession number DRX155515); GES, a clinical strain of *bla*_GES-24_ positive *K. pneumoniae* (accession number DRA008464); OXA-48, a clinical strain of *bla*_OXA-232_ positive *K. pneumoniae* (accession number DRX114774); and VIM, a clinical strain of *bla*_VIM-1_ positive *K. pneumoniae* (accession number SAMEA2709037). We also applied six CPE clinical isolates that were confirmed to carry both NDM (*bla*_NDM-1_) and OXA-48-like carbapenemase (*bla*_OXA-232 or 181_) genes by a multiplex PCR-based method [11] to confirm the utility of our assay to detect isolates harboring dual carbapenemase genes. We also applied 66 clinical isolates of Carbapenem Inactivation Method (CIM)-negative *Enterobacteriaceae* to evaluate the accuracy of the assay. The need for informed consent was waived because the study used only isolated pathogens and an individual’s information was anonymized.

#### 2.1.2. Primers and Probes

The details of primers and probes for NDM, KPC, IMP, GES, OXA, and VIM are shown in Table 1.

#### 2.1.3. PCR Conditions

In the beginning, we intended to detect all six carbapenemase genes in a single assay; however, the equipment can process only up to four fluorescent dyes simultaneously. Therefore, we constructed two multiplex assays to detect two combination sets of three carbapenemase genes and an internal control; Set 1 and Set 2 consisted of *bla*_NDM_, *bla*_KPC_, and *bla*_IMP_, and *bla*_GES_, *bla*_OXA-48_, and *bla*_VIM_, respectively.

A 20-μL reaction mixture consisting of 5 μL sample DNA and 15 μL master mix was used for this assay. For each sample, DNA was isolated by boiling bacterial suspension for 10 min. For each reaction, the master mix contained 4 μL LightCycler^®^ Multiplex DNA Master; 0.5 μL of each LightMix^®^ Modular series primer and probe (TIB Molbiol GmbH, Berlin, Germany; Table 1) and 0.5 μL of LightMix^®^ Modular PhHV internal control (2 μL in total) (TIB Molbiol GmbH, Berlin, Germany); 0.25 μL LightCycler^®^ Uracil-DNA Glycosylase (UNG) (Roche Diagnostics K.K., Tokyo, Japan); and 8.75 μL PCR-grade water (Thermo Fisher Scientific Inc., Japan). The following wavelengths were used: Set 1: NDM, FAM (Excitation [Ex]. 465 nm; Emission [Em]. 510 nm); KPC, LC610 (Ex. 540 nm; Em. 610 nm); IMP, LC640 (Ex. 610 nm; Em. 645 nm); and PhHV, LC670 (Ex. 610 nm; Em. 670 nm). Set 2: GES, LC640 (Ex. 610 nm; Em. 645 nm); OXA, LC580 (Ex. 540 nm; Em. 580 nm); VIM, Cyan500 (Ex. 465 nm; Em. 510 nm); and PhHV, LC670 (Ex. 610 nm; Em. 670 nm). Thermal cycling parameters were as follows: UNG activation at 40 °C for 10 min (1 cycle), Taq activation at 95 °C for 10 min (1 cycle), followed by 45 cycles at 95 °C for 15 s, 60 °C for 30 s, and 72 °C for 2 s (amplification), and 45 °C for 30 s (cooling). The total duration of this assay is approximately 60 min. We defined the detection limit as a concentration at which the coefficient of variation of the five replicates (Log) per tube (15 µL) is less than 10%. Using this assay, we first used the DNA standards of *bla*_NDM_, *bla*_KPC_, *bla*_OXA-48_, *bla*_IMP_, *bla*_VIM_, and *bla*_GES_ (LightMix^®^ Modular series; TIB Molbiol GmbH, Berlin, Germany) to construct the standard curves. Four-fold serial dilutions (8, 40, 200, 1000 copies/15 μL) of the positive controls were prepared and the experiments were performed in duplicate.

#### 2.1.4. Equipment

In this study, we used the COBAS^®^ z480 (Roche Diagnostics K.K., Tokyo, Japan) genetic analysis system.

## 3. Results

We constructed standard curves for the six carbapenemase genes using DNA standard solutions. As shown in Figure 1, the multiplex assays showed satisfactory accuracy for all genes. The correlation coefficients were greater than 0.99 for all genes. The lower limits of quantification were confirmed to be ≥20 copies/15 μL for *bla*_NDM_, *bla*_KPC_, *bla*_IMP_, *bla*_GES_, and *bla*_OXA-48_, and ≥10 copies/15 μL for *bla*_VIM_ (Appendix A). The multiplex assay detected each target gene of the reference or clinical strains without cross reactivity (Table 2). Finally, the assay detected two distinct carbapenemase genes (NDM and OXA-48-like carbapenemase) without false-positive results (Table 3). The assay did not show a false-positive result for the 66 CIM-negative isolates at all.

Four-fold serial dilutions of positive controls (LightMix^®^ Modular series, Roche Diagnostics K. K., Tokyo, Japan; 8, 40, 200, 1000 copies/15 μL) were used as standard solutions. A line indicates the linear detection range. Correlation coefficients were greater than 0.99 for all genes.

## 4. Discussion

In this study, we demonstrated a potential utility of the multiplex quantitative real-time PCR assay for six major carbapenemase genes using the clinically available COBAS^®^ z480 equipment. The major advantages of the assay are high sensitivity and short procedure duration (60 min). In addition, the use of UNG and internal control improves assay reliability by preventing carry-over contamination [12] and allowing visualization of amplification products. The wide availability of the equipment at medical facilities may also be beneficial for our assay. A recent study has also described a multiplex PCR assay using LightCycler^®^ 480 II (Roche Diagnostics KK, Tokyo, Japan) for the detection of carbapenemase genes [13]. However, the device is for research use only and is not clinically available. Lastly, the cost of our assay is approximately half of that for an existing cassette-type multiplex RT-PCR assay, such as the Xpert CARBA-R assay (approximately 3250 JPY versus 6000 JPY per run, respectively) [14]. Collectively, our method has several strengths compared to the previous methods [15,16,17,18].

A potential disadvantage of our assay is that it requires two sets-per-unit to differentiate between six carbapenemase genes; this can be technically complex. We have included *bla*_GES_ in the assay as one of the carbapenemases, although the majority of those may be classified into class A extended-spectrum beta-lactamases. Our aim of developing this assay was to widely screen for CPE, and thus, we incorporated *bla*_GES_ into the primer sets. In fact, many previous studies dealt with *bla*_GES_ as carbapenemase [19,20,21,22]. Additionally, an insufficient number of isolates were tested to estimate sensitivity and specificity of the assay, especially below the detection limit. We have no solid explanation for the reason of difference in the detection limit between *bla*_VIM_ and other variants. In comparison with other countries, the detection rates of CPE isolates in clinical situations are infrequent in Japan. It is, thus, hardly possible to increase the isolates to be tested for the study, and we intend to publish the present data as a preliminary study. Therefore, further investigations should be performed before clinical use. Furthermore, the utility of the assay to directly apply clinical specimen should also be examined.

The early detection of CPE can help in implementing effective countermeasures at an appropriate time to initiate early treatment interventions. The increasing global dissemination of CPE necessitates the preparation of a range of mobile carbapenemase genes. The plasmids harboring carbapenemase genes can be transmitted easily among patients. Our method using the clinically available laboratory equipment can search for genetically diverse CPE isolates with a high accuracy. We believe that our assay would be significantly useful in clinical laboratories.

## Figures and Tables

**Figure 1 pathogens-10-00276-f001:**
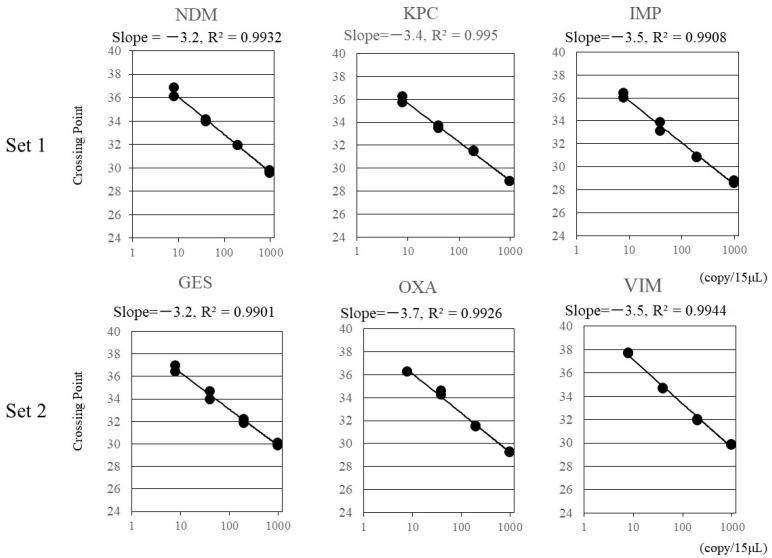
Standard curves for the four-fold serial dilutions of the DNA standards.

**Table 1 pathogens-10-00276-t001:** Primers and probes used in the multiplex, quantitative, real-time PCR test.

Gene	Primer Names	Probe	Sequences (5′-3′)	Wavelength (nm)
NDM	LightMix^®^ Modular NDM Carbapenemase	FAM labeled hydrolysis probe	F: GGTTTGGCGATCTGGTTTTC R: CGGAATGGCTCATCACGATC	FAM (465–510)
KPC	LightMix^®^ Modular KPC Carbapenemase	LC610 labeled hydrolysis probe	F: ATGTCACTGTATCGCCGTCT R: TTTTCAGAGCCTTACTGCCC	610 (540–610)
IMP	LightMix^®^ Modular IMP Carbapenemase	LC640 labeled probe	F: GAATAGRRTGGCTTAAYTCTC R: CCAAACYACTASGTTATC	640 (610–645)
GES	LightMix^®^ Modular GES Carbapenemase	LC640 labeled probe	F: GCTTCATTCACGCACTATT R: CGATGCTAGAAACCGCTC	640 (610–645)
OXA	LightMix^®^ Modular OXA-48 Carbapenemase	R6G labeled hydrolysis probe	F: TTGGTGGCATCGATTATCGG R: GAGCACTTCTTTTGTGATGGC	580 (540–580)
VIM	LightMix^®^ Modular VIM Carbapenemase	Cyan500 labeled hydrolysis probe	F: GTTTGGTCGCATATCGCAAC R: AATGCGCAGCACCAGGATAG	FAM (465–510)

**Table 2 pathogens-10-00276-t002:** Multiplex quantitative real-time PCR assay for six *Enterobacteriaceae* isolates harboring distinct carbapenemases.

PCR Set	Carbapenemase Type	Genotypes of the Tested Isolates					
		*bla* _NDM-5_	*bla* _KPC-2_	*bla* _IMP-6_	*bla* _GES-24_	*bla* _OXA-232_	*bla* _VIM-1_
1	NDM	19.52/19.36	n.d.	n.d.	n.d.	n.d.	n.d.
	KPC	n.d.	18.33/18.46	n.d.	n.d.	n.d.	n.d.
	IMP	n.d.	n.d.	15.93/15.85	n.d.	n.d.	n.d.
2	GES	n.d.	n.d.	n.d.	16.13/16.16	n.d.	n.d.
	OXA	n.d.	n.d.	n.d.	n.d.	22.11/22/15	n.d.
	VIM	n.d.	n.d.	n.d.	n.d.	n.d.	17.67/17.62

Target genes for PCR set 1 are *bla*_NDM_, *bla*_KPC_, and *bla*_IMP_, and target genes for set 2 are *bla*_GES_, *bla*_OXA_, and *bla*_VIM_. Shown are crossing points (duplicate). n.d., not detected.

**Table 3 pathogens-10-00276-t003:** Multiplex quantitative real-time PCR assay for six *Enterobacteriaceae* clinical isolates carrying both NDM andOXA-48 like genes.

PCR Set	Carbapenemase Type	Genotypes of the Tested Isolates					
		Isolate 1 *bla*_NDM-1_ + *bla*_OXA232_	Isolate 2 *bla*_NDM-1_ + *bla*_OXA232_	Isolate 3 *bla*_NDM-1_ + *bla*_OXA232_	Isolate 4 *bla*_NDM-1_ + *bla*_OXA181_	Isolate 5 *bla*_NDM-1_ + *bla*_OXA232 or 181_	Isolate 6 *bla*_NDM-1_ + *bla*_OXA232 or 181_
1	NDM	26.63/26.63	25.95/25.87	26.21/26.16	17.78/17.76	27.07/27.1	25.44/25.54
	KPC	n.d.	n.d.	n.d.	n.d.	n.d.	n.d.
	IMP	n.d.	n.d.	n.d.	n.d.	n.d.	n.d.
2	GES	n.d.	n.d.	n.d.	n.d.	n.d.	n.d.
	OXA	27.63/27.85	26.95/26.94	27.34/27.31	25.95/25.72	29.55/29.56	30.42/30.2
	VIM	n.d.	n.d.	n.d.	n.d.	n.d.	n.d.

Shown are threshold cycles (duplicate). n.d., not detected. All the tested isolates were confirmed to harbor both *bla*_NDM_ and *bla*_OXA_, but not other types such as *bla*_KPC_, *bla*_IMP_, *bla*_GES_, and *bla*_VIM_, by a previously reported method [11].

## Data Availability

The data presented in this study are available on request from the corresponding author. The data are not publicly available because they are not opened.

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
