# Peer review of "Multiplex Real-Time PCR Assay for Six Major Carbapenemase Genes"

_pathogens, 2021, doi:10.3390/pathogens10030276_

Round 1

Reviewer 1 Report

This revised version of the manuscript does not show significant improvements.

Still problems remain the same; 

  • many other similar assays have already been published
  • the number of isolates tested is by far insufficient
  • detecting "GES" encoding genes as a whole is meaningless, considering that only few of those GES variants actually possess carbapenemase activities; the others are classical ESBLs. Thus the assay has no value if not able to differentiate both types of enzymes
  • Despite OXA-48 is by far the most common OXA-48-like enzyme worldwide, no isolate producing this variant has been tested !

Author Response

Comments from the reviewer

Still problems remain the same; 

Response to the Reviewer

Thank you again for your suggestion.

Comment #1

Many other similar assays have already been published

Response #1

As indicated, literature has shown that there are several other assays to be used for CPE detection. However, as noted in the manuscript, our method has many advantages to be addressed, including high sensitivity, short procedure duration (60 min), reliability by preventing carry-over contamination, visualization of amplification products, wide availability of the equipment at medical facilities, and lower cost. Please see the 1st paragraph of the Discussion.

Comment #2

The number of isolates tested is by far insufficient

Response #2

Because we have fewer cases of CPE in Japan, it is actually difficult to gather more isolates to be tested. We admit that we need to perform a further investigation before correctly evaluating the utility of our assay.

Comment #3

Detecting "GES" encoding genes as a whole is meaningless, considering that only few of those GES variants actually possess carbapenemase activities; the others are classical ESBLs. Thus the assay has no value if not able to differentiate both types of enzymes

Response #3

Majority of GES may be classified into ESBLs but not carbapenemases, as suggested. However, many previous studies dealt with GES as carbapenemase (PMID: 33008738, 30538682, 32475870, 30189011, 27716102 and so on), including GES-24 (PMID: 30289383, 27671068, 31800356). Our aim of developing this method was to screen CPE isolates, and therefore, we included GES variants. When GES was found to be positive using our test, it is necessary to confirm the activity of carbapenemase by sequencing or other methods.

Comment #4

Despite OXA-48 is by far the most common OXA-48-like enzyme worldwide, no isolate producing this variant has been tested !

Response #4

We targeted the OXA-48 variants as well. Please go through the manuscript as a whole.

Thank you for your review.

Reviewer 2 Report

Dear Authors

The work by Yoshioka et al,  entitled Multiplex Real-Time PCR Assay for Six Major Carbapenemase six  Genes.

Please see my comments

Regards

Mez

Author Response

Comments to Reviewer 2

Thank you for your review. We have revised our manuscript according to your valuable comments.

Major changes and comments are given below;

  • As suggested, we have restructured the Materials and Methods section by Bacterial strains, Primers and probes, PCR conditions, and Equipment.
  • We have designed the primers with a help of TIB-MOLBIOL. Com
  • We have not confirmed the sensitivity of our assay below the detection limit in comparison with conventional PCR assay.
  • We have no solid explanation for the reason of difference in the detection limit between blaVIM and other bariants.
  • Crossing points indicated “Threshold Cycle, Ct)”. We have changed the term.
  • We selected the COBAS® z480 equipment because that is a globally-available machine to be used in clinical- or hospital laboratory.

Reviewer 3 Report

These following concerns need to be addressed:

The Authors should inform about source of sequences of genes and design of primers. Moreover, the Authors should write about analytical sensitivity, specificity linearity and reproducibility of the assay.

How did the authors get the bacterial suspension?

Procedure of DNA isolation should be more detailed / clarified.

How were the DNA samples eluted?

Did the authors estimation of purity of the extracted DNA?

How many final concentrations of each primer were contained in the reaction mixture?

What was the negative control?

Which software did the Authors use to statistical analysis and calculation?

The discussion is insufficient compared to other articles. The Authors should refer to previous studies in this same range, i.e.

  1. Jussimara Monteiro, Raymond H. Widen, Antonio C. C. Pignatari, Carly Kubasek, Suzane Silbert. Rapid detection of carbapenemase genes by multiplex real-time PCR. J Antimicrob Chemother 2012, https://doi.org/10.1093/jac/dkr563
  2. Anneke van der Zee, Lieuwe Roorda, Gerda Bosman, Ad C Fluit, Mirjam Hermans, Paul HM Smits, Adri GM van der Zanden, René te Witt, Lesla ES Bruijnesteijn van Coppenraet, James Cohen Stuart. Jacobus M Ossewaarde. Multi-centre evaluation of real-time multiplex PCR for detection of carbapenemase genes OXA-48, VIM, IMP, NDM and KPC. BMC Infectious Diseases 2014;14, Article number: 27.
  3. Smiljanic M., Kaase M., Ahmad-Nejad P., Ghebremedhin Comparison of in-house and commercial real time-PCR based carbapenemase gene detection methods in Enterobacteriaceaeand non-fermenting gram-negative bacterial isolates. Annals of Clinical Microbiology and Antimicrobials  2017;16, Article number: 48, https://doi.org/10.1186/s12941-017-0223-z
  4. Jeanette W.P. Teo, My-Van La, Raymond T. P. Lin. Development and evaluation of a multiplex real-time PCR for the detection of IMP, VIM, and OXA-23 carbapenemase gene families on the BD MAX open system. Diagnostic Microbiology and Infectious Disease 2016;86(4): 358-361, https://doi.org/10.1016/j.diagmicrobio.2016.08.019

Author Response

Comment from Reviewer

These following concerns need to be addressed:

Response

Thank you for your review.

Comment #1

The authors should inform about source of sequences of genes and design of primers. Moreover, the authors should write about analytical sensitivity, specificity linearity and reproducibility of the assay.

Response #1

Thank you for your suggestion. The source of the gene sequence and primer design are described in Table 1. All the analytical sensitivity, specificity linearity and reproducibility of the assay is summarized in the first paragraph of the Results section (See Figure 1, Supplemental Table 1 as well)

Comment #2

How did the authors get the bacterial suspension?

Response #2

As described in the Material and Methods, we used both clinical isolates and reference isolates. For the detail of reference isolates, information is given in the “Bacterial strains” section.

Comment #3

Procedure of DNA isolation should be more detailed / clarified. How were the DNA samples eluted?

Response #3

Thank you for your question. We heated the bacterial suspension in an incubator for 10 minutes and spin it down. Details are given in the “PCR conditions” section of the Material and Methods.

Comment #4

Did the authors estimation of purity of the extracted DNA?

Response #4

In this study, we have not assessed the purity of the extracted DNA.

Comment #5

How many final concentrations of each primer were contained in the reaction mixture?

Response #5

It contains 0.5µL per tube (20µL).

Comment #6

What was the negative control?

Response #6

We used grade water for the negative control.

Comment #7

Which software did the Authors use to statistical analysis and calculation?

Response #7

We used the Office software Excel.

Comment #8

The discussion is insufficient compared to other articles. The Authors should refer to previous studies in this same range, i.e.

  1. Jussimara Monteiro, Raymond H. Widen, Antonio C. C. Pignatari, Carly Kubasek, Suzane Silbert. Rapid detection of carbapenemase genes by multiplex real-time PCR. J Antimicrob Chemother2012, https://doi.org/10.1093/jac/dkr563
  2. Anneke van der Zee, Lieuwe Roorda, Gerda Bosman, Ad C Fluit, Mirjam Hermans, Paul HM Smits, Adri GM van der Zanden, René te Witt, Lesla ES Bruijnesteijn van Coppenraet, James Cohen Stuart. Jacobus M Ossewaarde. Multi-centre evaluation of real-time multiplex PCR for detection of carbapenemase genes OXA-48, VIM, IMP, NDM and KPC. BMC Infectious Diseases2014;14, Article number: 27.
  3. Smiljanic M., Kaase M., Ahmad-Nejad P., Ghebremedhin Comparison of in-house and commercial real time-PCR based carbapenemase gene detection methods in Enterobacteriaceaeand non-fermenting gram-negative bacterial isolates. Annals of Clinical Microbiology and Antimicrobials2017;16, Article number: 48, https://doi.org/10.1186/s12941-017-0223-z
  4. Jeanette W.P. Teo, My-Van La, Raymond T. P. LinDevelopment and evaluation of a multiplex real-time PCR for the detection of IMP, VIM, and OXA-23 carbapenemase gene families on the BD MAX open system. Diagnostic Microbiology and Infectious Disease 2016;86(4): 358-361, https://doi.org/10.1016/j.diagmicrobio.2016.08.019

Response #8

Thank you for your suggestion. We have checked the recommended literature. And as a result, we found that…

  • Reference 1. is similar to our method, but the measurement time is longer than our method.
  • Reference 2. does not include GES, which is present in our method.
  • Reference 3. does not include the IMP and GES.
  • Reference 4 does not include NDM, KPC and GES in our method.

We added these references in the manuscript (Discussion section)

Thank you for your review.

Round 2

Reviewer 1 Report

Still same comments than previously. Nothing to add.

Author Response

Reviewer 1

Still same comments than previously. Nothing to add.

Comment

Thank you very much for your repeated check.

Again, we explain our opinion to your previous comments.

  • You have indicated that there already have been similar assays. However, our method has many advantages to be addressed, including high sensitivity, short procedure duration (60 min), reliability by preventing carry-over contamination, visualization of amplification products, wide availability of the equipment at medical facilities, and lower cost. Please again refer to the 1st paragraph of the Discussion.

  • The number of tested isolates is insufficient, as indicated. This is just because we have fewer cases of CPE in Japan. As described in the literature, blaIMP CPE is dominantly prevailing in Japanese clinical settings and it is hardly possible to detect other genetic types of CPE. We understand this limitation of the study, and thus described it in the limitation section (2nd paragraph of the Discussion).

Despite these limitations, we believe our manuscript is of value to be published in the literature. We hope your sincere understanding.

Reviewer 2 Report

Dear Authors,

Please include the following as limitations of the present study after the first paragraph of the discussion" Collectively, our method has several
138 strengths to be addressed, compared to the previous methods [15-18].

  • We have not confirmed the sensitivity of our assay below the detection limit in comparison with conventional PCR assay.
  • We have no solid explanation for the reason of difference in the detection limit between blaVIM and other variants.

Correct TIB-MOLBIOL. com in the text, add proper address!

Pleas indicate the ethical approval for the clinical samples, their origin, and if any data available about these clinical isolates used in your assay. 

Best

Mez

Author Response

Reviewer 2

Please include the following as limitations of the present study after the first paragraph of the discussion" Collectively, our method has several strengths to be addressed, compared to the previous methods [15-18].

  • We have not confirmed the sensitivity of our assay below the detection limit in comparison with conventional PCR assay.
  • We have no solid explanation for the reason of difference in the detection limit between blaVIMand other variants.

Comment

Thank you for your suggestion. We have provided the limitation sentences in the second paragraph of the discussion (Line 137-144).  

  • Correct TIB-MOLBIOL. com in the text, add proper address!

Comment

We have corrected the address of the company in the method section.

  • Pleas indicate the ethical approval for the clinical samples, their origin, and if any data available about these clinical isolates used in your assay. 

Comment

Our IRB committee approved that we do not need obtain informed consent from each patient because of the anonymized method. We have added the following sentence in the Bacterial strains section; “A need of informed consent was waived because the study uses only isolated pathogens and individual’s information is anonymized.”

Thank you for your review.

Reviewer 3 Report

Authors have revised the manuscript taking into account my suggestions and comments.

Author Response

Reviewer 3

Authors have revised the manuscript taking into account my suggestions and comments.

Comments

Thank you for your valuable comments for our manuscript. We appreciate your contribution.

This manuscript is a resubmission of an earlier submission. The following is a list of the peer review reports and author responses from that submission.

Round 1

Reviewer 1 Report

This is another molecular assay to detect carbapenemase encoding genes, among many others previously published. Therefore it lacks originality. 

The number of isolates tested is minimal, and the procedure is poorly explained. Is that test supposed to be used from isolated colonies, or can it be used from clinical samples?

The design of the study itself is questionable, with GES encoding genes being included. However, most of the GES enzymes are NOT carbapenemases, only few variants are. Therefore the specificity of the technique is questioned here. How to recognise whether this is an ESBL (such as GES-1) or a carbapenemase (such as GES-5) using your technique, if sequencing is not performed?